# Peri-Neural Invasion Is an Important Prognostic Factor of T2N0 Oral Cancer

**DOI:** 10.3390/medicina58121809

**Published:** 2022-12-08

**Authors:** Chi-Sheng Cheng, Chien-Chih Chen, Yi-Chun Liu, Chen-Chi Wang, Yu-Shu Chou

**Affiliations:** 1Department of Oral and Maxillofacial Surgery, Taichung Veterans General Hospital, Taichung 407219, Taiwan, China; 2Department of Radiation Oncology, Taichung Veterans General Hospital, Taichung 407219, Taiwan, China; 3Department of Medical Imaging and Radiological Sciences, Central Taiwan University of Science and Technology, Taichung 406053, Taiwan, China; 4Institute of Clinical Medicine, National Yang Ming University, Taipei 407219, Taiwan, China; 5Department of Audiology and Speech-Language Pathology, Asia University, Taichung 407219, Taiwan, China; 6School of Medicine, National Yang-Ming University, Taipei 407219, Taiwan, China; 7Department of Otolaryngology Head and Neck Surgery, Taichung Veterans General Hospital, Taichung 407219, Taiwan, China; 8Department of Nuring, Taichung Veterans General Hospital, Taichung 407219, Taiwan, China

**Keywords:** adjuvant therapy, angiolymphatic invasion, chemotherapy, perineural invasion, oral cancer, radiotherapy, squamous cell carcinoma

## Abstract

*Background and objectives*: Among patients with pathologically proven T2N0 oral squamous cell carcinoma (OSCC), a notable amount of patients still die from tumor recurrence although they have radical surgery for early stage cancers. In literature, the prognostic indicators of this specific disease entity were rarely reported. This study aims at analyzing the prognostic factors of T2N0 OSCC patients and discussing possible managements to improve the survival. Materials and Methods: From January 2012 to December 2017, the data of 166 pathologically proven T2N0 oral cancer patients proved by radical surgery were retrospectively collected. The clinical and pathologic factors including age, gender, tumor differentiation grade, perineural invasion (PNI), angiolymphatic invasion (ALI), margin status, and adjuvant therapy were analyzed by univariate and multivariate analysis to determine their association with disease-specific survival (DSS), and disease-free survival (DFS), which were calculated by Kaplan–Meier method. Results: After median follow up time of 43.5 months, overall 3-year rates of DSS and DFS were 86.1% and 80.1% respectively for our 166 patients. Univariate analysis showed that the 3-year DSS of 90.8% for PNI negative patients was significantly better than DSS of 57.0% for PNI positive patients (*p* = 0.0006). The 3-year DFS of 84.2% for PNI negative patients was also significantly better than DFS of 54.6% for PNI positive patients (*p* = 0.001). Further multivariate analysis revealed PNI was the only independent prognostic factor associated with both DSS (Hazard Ratio (HR) = 5.02; 95% Confidence Interval (CI) = 1.99–12.6; *p* = 0.001), and DFS (HR = 3.92; 95% CI = 1.65–9.32; *p* = 0.002). Nearly 10% (16) of the 166 patients had adverse pathologic feature of PNI only. In the 11 patients without adjuvant therapy, 5 patients died from OSCC. No patients had recurrence or mortality after they received adjuvant therapy with chemotherapy ± radiotherapy. Conclusion: PNI was an independent prognostic factor for T2N0 oral cancer patients. Adjuvant chemotherapy and radiotherapy may benefit the survival of this specific disease entity, but further investigations are needed to elucidate the optimal regimen.

## 1. Introduction

Oral squamous cell carcinoma (OSCC) is one of the common malignancies worldwide [1]. Surgery is the mainstay treatment for OSCC, particularly for early-stage disease. Locoregional recurrence occurs frequently after radical surgery, particularly for patients with positive surgical margins and extranodal extension in cervical metastasis [2,3]. Postoperative adjuvant therapy has been recommended for advanced OSCC patients who have aforementioned pathological adverse features [2,3,4]. For pathologically proven T2N0 OSCC patients, the margins positive rate should not be high because of the limited size of early stage tumors. In addition, there is no cervical metastasis. It indicates that they might only need resection of tumor and elective neck dissection. However, previous studies [5,6] have revealed that the local recurrence rate was still as high as 23% after 31 months follow up and the 5-year survival rate was only 63.5% for T2N0 OSCC. These clinical reports suggest that some pathological adverse features and occult diseases may affect clinical outcomes in T2N0 OSCC patients. Because the rate of salvage surgery for recurrent disease is poor [7,8], it is important to identify the potential risk factors for disease recurrence and give the patient possible adjuvant therapy to reduce the risks. 

Peri-neural invasion (PNI) and angio-lymphatic invasion (ALI) are two adverse pathologic features in many human malignancies including OSCC [9,10]. Chatzistefanou [9] reviewed previous articles and concluded that PNI is correlated with more aggressive tumor and poor outcomes. They are also required elements in a standard pathology report for OSCC [11]. However, the role of PNI and ALI were not clear yet. When patient has only one risk factor of PNI or ALI, the need for adjuvant therapy still remains controversial. Therefore, we attempted to review the data of our patients of T2N0 OSCC to elucidate the risk factors of poor prognosis and to suggest possible management to improve the patient’s survival.

## 2. Materials and Methods

### 2.1. Study Subjects Inclusion Criteria

This study included 287 newly diagnosed OSCC patients from January 2012 to December 2017 in the cancer registry database of Taichung Veterans General Hospital. The inclusion criteria for the study were: (1) patients who had completed pretreatment cancer staging workup, including complete blood cell count and differential count, liver function test, renal function test, computed tomography or magnetic resonance imaging, chest X ray, whole body bone scan, and abdomen ultrasonography; (2) patients who had no second primary malignancy or distant metastasis at diagnosis; (3) patients who had received radical tumor excision and neck dissection with pathology proved tumor size of 2–4 cm (T2) and no lymph node metastasis (N0) according to the American Joint Committee on Cancer (AJCC) TNM classification system 7th edition. All participants provided written informed consent to receive the surgery and the Institutional Review Board of Taichung Veterans General Hospital approved this retrospective study (protocol number CE19242A). Afterwards, there were 166 patients eligible for this study according to the aforementioned criteria.

### 2.2. Adjuvant Therapy after Surgery

Patients with pathologic risk factors including positive margin, PNI, ALI, and tumor cell poor differentiation chose their adjuvant therapy such as radiotherapy, chemotherapy or both after knowing the pros and cons of the treatments. If patients refused adjuvant therapy, then close surveillance was done. A total of 31 patients underwent external beam radiotherapy using a linear accelerator with a 6-MV photon beam and source-axis distance technique. A total radiation dose of 60.0–66.0 Gy, 1.8–2.0 Gy per fraction, at 5 fractions per week was delivered. Of the 31 patients, 25 were scheduled to receive concurrent cisplatin-base chemotherapy. The regimen was (1) cisplatin 20 mg/m^2^ and 5FU 400 mg/m^2^ for 1–4 days, every 4 weeks; or (2) weekly cisplatin 30–50 mg/m^2^; or (3) tri-weekly cisplatin 100 mg/m^2^. The adjuvant therapy was delivered within 4–6 weeks after radical surgery. Any mortality caused by the therapy or disease was calculated as disease-specific mortality.

### 2.3. Statistical Analysis

The endpoints were disease-specific survival (DSS) and disease-free survival (DFS). DSS was calculated from the date of radical surgery to the date of disease-related death or last follow-up. DFS was measured from the date of radical surgery to the date of any evidence of recurrence or last follow-up. Survival times were estimated using the Kaplan–Meier method, while the Log-rank test was used for the comparison between the groups. A Cox regression model was used for multivariate analysis. The statistical analyses were performed using SPSS software (version 12.0, IBM, New York, NY, USA). A *p*-value of less than 0.05 was considered statistically significant.

## 3. Results

There were 148 males and 18 females with age ranging from 33 to 92 years old with median age of 56 years. Patients were dichotomized to old age group (>56 years old) and young age group (≤56 years old). Histopathologic examinations revealed that 20 (12%) patients had positive margins, 24 (14.4%) patients had PNI, but only 6 (3.6%) patients had ALI. A total of 28 (16.8%) patients received adjuvant radiotherapy, while 24 (14.4%) patients received adjuvant chemotherapy.

After, follow-up time ranged from 2.3 to 101.6 months with median of 43.5 months, the 3-year rates of DSS and DFS were 86.1% and 80.1%, respectively. The patients’ characteristics and the results of univariate analysis are summarized in Table 1. Only factors having significant association with survival are mentioned in the following text. The 3-year DSS rates for PNI positive and PNI negative were 57.0% and 90.8%, respectively (*p* = 0.0006, Figure 1); the 3-year DFS rates for PNI positive and PNI negative were 54.6% and 84.2% respectively (*p* = 0.001, Figure 2). The 3-year DSS rates for ALI positive and ALI negative were 50.0% and 87.5%, respectively (*p* = 0.01). The 3-year DFS rates for ALI positive and ALI negative were 50.0% and 82.2%, respectively (*p* = 0.0009). The 3-year DSS rates for patients who had received adjuvant chemotherapy and those who had not were 100% vs. 82.4% (*p* = 0.01). The 3-year DFS rates for patients who had received adjuvant chemotherapy and those who had not were 100% vs. 76.7% (*p* = 0.02).

The significant factors in aforementioned univariate analysis were incorporated into the multivariate analysis and the results are summarized in Table 2. It revealed that PNI (Hazard Ratio (HR) = 5.02; 95% Confidence Interval (CI) = 1.99–12.6; *p* = 0.001) was an independent prognostic factor for DSS. PNI (HR = 3.92; 95% CI = 1.65–9.32; *p* = 0.002) and adjuvant chemotherapy (HR = 0.06; 95% CI = 0.01–0.52; *p* = 0.010) were independent prognostic factors for DFS.

In Table 1, 24 patients had PNI. However, after excluding PNI positive patients with other simultaneous adverse features, there were 16 patients who had PNI as the only adverse pathologic feature. Among the 16 patients, 5 patients received adjuvant therapy (4 with radiotherapy with chemotherapy, 1 with chemotherapy alone). The other 11 patients chose close surveillance. Patients who received adjuvant therapy had no recurrence in the follow up and no mortality caused by the disease or its treatment complications. Conversely, for 11 patients without adjuvant therapy, 3 patients had primary tumor recurrence, 1 patient had neck lymph nodes recurrence, and 2 patients had both primary and regional recurrence. Five patients without adjuvant therapy died of disease in the follow up period. Although the number of this PNI positive cohort was small and no statistical results could be concluded, the Kaplan–Meier curves of DSS and DFS are shown in Figure 3 and Figure 4 for readers’ reference. It can be deduced that adjuvant therapy may largely reduce disease recurrence and mortality in our observation. Compared to PNI, 5 of our 6 ALI positive patients also had simultaneous PNI. Therefore, the rate of only ALI was only 0.6% (1/166) and it needs no further discussion.

## 4. Discussion

T2N0 OSCC is an early-stage disease; however, the recurrence rate and mortality rate are not neglectable according to our results and the literature reports. Therefore, it is important to identify the T2N0 OSCC patients with poorer outcomes and give them more aggressive management than surgery alone to improve the survival.

PNI has been widely considered to be an adverse feature in histopathology, which is related to aggressive tumor behavior in oral cancer patients [12,13,14,15]. PNI is a way for tumor cell spread to occur in and along the nerve bundle beyond the local tumor. In a previous study, OSCC was recognized as a neurotropic cancer, and approximately 30% of the patients had PNI positive [16]. In fact, PNI positive rates are 2-fold and 3-fold higher in patients with N positive and bigger primary T3 to T4 tumors. Post-operative adjuvant therapy was already recommended for late-stage disease according to present treatment policy. Conversely, the role of PNI in pathologically proven T2N0 patients can be more contributory to guide the postoperative adjuvant therapy in early-stage disease. In this study, although only 14.4% of our patients had positive PNI, the positive rate was even higher than margin positive rate (12%). In the study by Tai et al. [17] on patients with primary T1-2 OSCC, the rate of PNI positive (27.4%) was also higher than the rate of margin positive (7.2%). Our observation was in compliance with their results. Furthermore, PNI is the only factor which was significantly associated with poor DSS and DFS in both univariate and multivariate analysis in our research. It revealed that, in early T2 OSCC with less difficulty of radical excision, PNI may play an important role to decide the surgical outcomes.

Angiolymphatic invasion or ALI is another adverse pathologic feature of OSCC. Sparano et al. [18] showed that ALI was a predictor correlated to occult lymph node metastasis in clinical T1/T2N0 oral cancer patients. However, occult cervical metastasis had been excluded by elective neck dissection in our study cohort. Therefore, the chance of ALI in our patients with pathologic N0 should be very low and our 3.6% rate of ALI positive confirmed this hypothesis. Although Aires et al. [19] found that ALI was a predictor for distant metastasis and our univariate analysis demonstrated that ALI positive patients had poorer DSS, and DFS. The very small rate of ALI positive in our T2N0 cohort did not influence survival in multivariate analysis. Furthermore, 5 of our 6 ALI positive patients also had simultaneous PNI. Therefore, only ALI positive needs no special attention.

For early-stage OSCC patients, the survival benefit of adjuvant therapy is controversial. Luryi et al. [20] reviewed 6830 stage I and II OSCC patients and revealed that neck dissection and treatment at academic institutes were associated with improved survival, because patients were more likely to receive neck dissection and less likely to have positive margins in research institutes. The study confirmed the importance of eliminating patients with occult nodal disease and their data showed that adjuvant therapy with radiotherapy or chemotherapy did not improve survival while neck dissection (48%) was not definitely performed. In the study on stage II tongue cancer by Rubin et al. [21], there was no survival benefit for patients who received adjuvant radiotherapy. However, their study did not consider the pathologic adverse factors such as PNI and ALI and our study may provide more information on the management of pathologic T2N0 OSCC patients with PNI positive. In the study on all stages of OSCC by Nair et al. [22], they concluded that aggressive treatment of the primary cancer with the coincident management of the neck is important in the presence of PNI. The PNI worsens survival and warrants intensification of adjuvant treatment. Our study results support their conclusion in T2N0 patients. Li et al. [23] conducted a systemic review and meta-analysis and demonstrated PNI significantly affected the locoregional recurrence and survival outcomes among oral tongue SCC patients. In our 16 patients with only PNI positive, patients had better outcome if adjuvant therapy was given.

It is difficult to draw a conclusion on what kind of adjuvant therapy must be given in patients with only PNI positive according to our retrospective study design. However, in the study by Rahima et al. [24], they found that PNI was significantly associated with regional recurrence and distant metastasis but not with local recurrence by multivariate analysis. According to their conclusion, chemotherapy should be considered for preventing distant metastasis. In our 5 patients receiving adjuvant chemotherapy (4 patients combined with radiotherapy), there was no treatment-related mortality. On the other hand, there were 5 local recurrences in our 11 only PNI positive patients without adjuvant therapy. Therefore, if there was no contraindication, we suggest chemotherapy in combination with radiotherapy for patients with PNI positive at present time.

This retrospective study did have other limitations. First, margin status did not show any significant difference in DSS and DFS. We reviewed the pathology reports and found true margin status may often be affected by different surgical procedures and further re-excision in the surgery process if margins were doubtful in frozen section during operation. Second, the staging was according to the AJCC 7th edition, and did not consider tumor invasion depth, which is considered in the new AJCC 8th edition. Third, we could not achieve a statistically significant conclusion by using the data of 16 patients with only PNI positive. However, we would like to report our important observation results on the poor survival effect of PNI positive. Hopefully, our study may invite further randomized control trials to elucidate the optimal management for early stage T2N0 patients with PNI.

## 5. Conclusions

For T2N0 OSCC patients, the pathologic adverse feature of PNI is not a rare finding. It is associated with poor DSS and DFS. Adjuvant chemotherapy and radiotherapy may benefit the survival of this specific disease entity, but further investigations are needed to elucidate the optimal regimen.

## Figures and Tables

**Figure 1 medicina-58-01809-f001:**
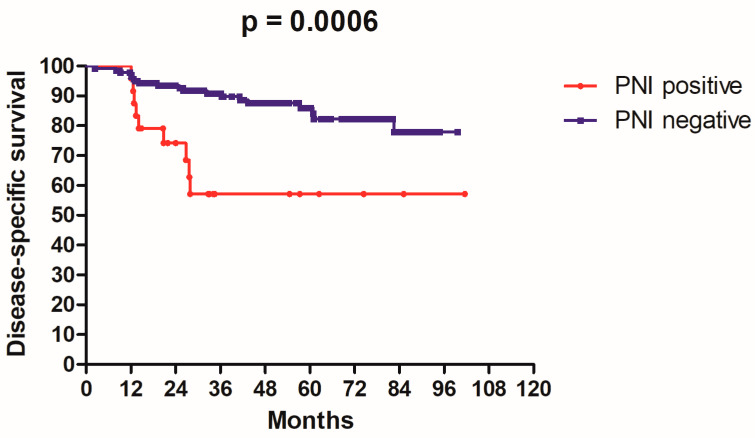
Kaplan–Meier disease-specific survival curve according to peri-neural invasion status in 166 patients with T2N0 OSCC.

**Figure 2 medicina-58-01809-f002:**
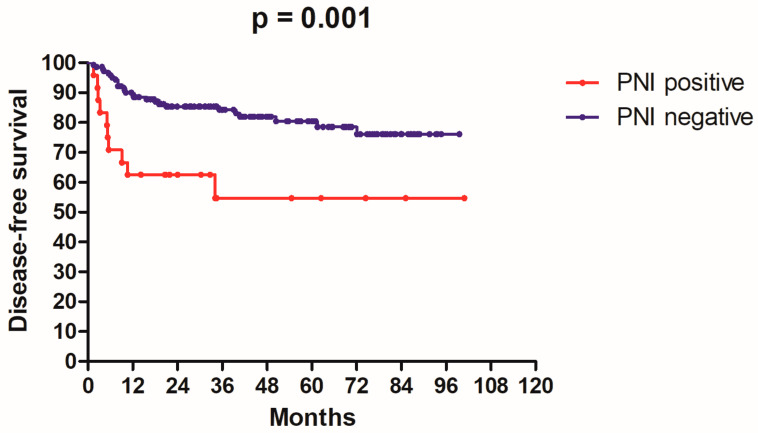
Kaplan–Meier disease-free survival curve according to peri-neural invasion status in 166 patients with T2N0 OSCC.

**Figure 3 medicina-58-01809-f003:**
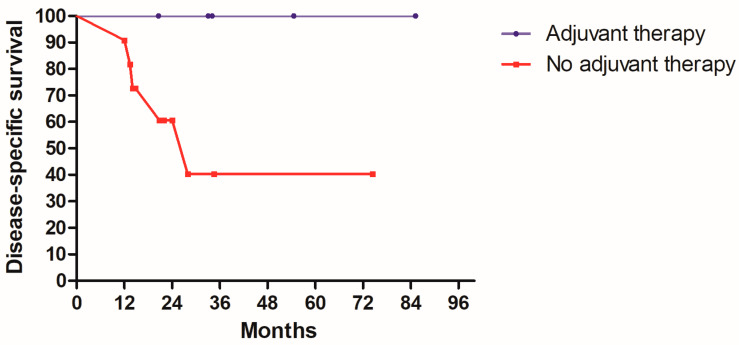
Kaplan–Meier disease-specific survival curve according to adjuvant therapy status in 16 patients with only PNI positive.

**Figure 4 medicina-58-01809-f004:**
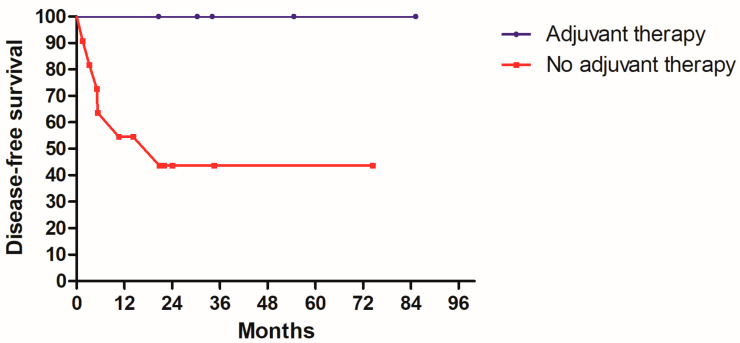
Kaplan–Meier disease-free survival curve according to adjuvant therapy status in 16 patients with only PNI positive.

**Table 1 medicina-58-01809-t001:** Patients’ characteristics and univariate analysis for disease-specific survival and disease-free survival in 166 patients with T2N0 OSCC.

Characteristics	Numbers (%)	3-Year DSS	*p* Value	3-Year DFS	*p* Value
Age	33–92				
	Median:56				
≤56	83 (50)	85.7%		86.4%	
>56	83 (50)	84.6%	*p* = 0.22	74.0%	*p* = 0.25
Gender					
Male	148 (89.2)	85.7%		80.1%	
Female	18 (10.8)	88.8%	*p* = 0.56	80.4%	*p* = 0.85
Grade					
Well differentiated	14 (8.4)	100%		84.4%	
Moderate differentiated	132 (79.6)	85.8%		81.2%	
Poor differentiated	20 (12.0)	70.1%	*p* = 0.56	70.0%	*p* = 0.53
PNI					
Positive	24 (14.4)	57.0%		54.6%	
Negative	142 (85.6)	90.8%	*p* = 0.0006	84.2%	*p* = 0.001
ALI					
Positive	6 (3.6)	50.0%		50.0%	
Negative	160 (96.4)	87.5%	*p* = 0.01	82.2%	*p* = 0.0009
Margin					
Positive	20 (12.0)	72.3%		68.8%	
Negative	146 (88.0)	86.9%	*p* = 0.25	81.6%	*p* = 0.33
Adjuvant radiotherapy					
Yes	28 (16.8)	88.2%		92.8%	
No	138 (83.2)	83.1%	*p* = 0.24	77.5%	*p* = 0.11
Adjuvant chemotherapy					
Yes	24 (14.4)	100%		100%	
No	142 (85.6)	82.4%	*p* = 0.01	76.7%	*p* = 0.02

Abbreviations: PNI: perineural invasion; ALI: angiolymphatic invasion; DSS: disease-specific survival; DFS: disease-free survival.

**Table 2 medicina-58-01809-t002:** Multivariate analysis for disease-specific survival and disease-free survival in 166 patients with T2N0 OSCC.

Variables	HR (95%CI)	*p* Value
DSS		
PNI	5.02 (1.99–12.6)	*p* = 0.001
ALI	1.79 (0.45–7.16)	*p* = 0.405
Adjuvant chemotherapy	0.01 (0.00–5.57)	*p* = 0.957
DFS		
PNI	3.92 (1.65–9.32)	*p* = 0.002
ALI	3.23 (0.95–10.9)	*p* = 0.059
Adjuvant chemotherapy	0.06 (0.01–0.52)	*p* = 0.010

Abbreviations: DSS: disease-specific survival; DFS: disease-free survival; PNI: perineural invasion; ALI: angiolymphatic invasion.

## Data Availability

All available data was included in this article.

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
