# Peer review of "Peri-Neural Invasion Is an Important Prognostic Factor of T2N0 Oral Cancer"

_medicina, 2022, doi:10.3390/medicina58121809_

Round 1

Reviewer 1 Report

The main question was to review the data of patients of T2N0 OSCC to elucidate the risk factors of poor prognosis and to suggest possible management to improve the patient’s survival. The topic of originality is excellent and relevant in the field. The main question posed the evidence and arguments presented, the references are appropriate, and the tables and figures are also appropriate. 

However, there are some questions and comments, please check and modify if necessary.

Thank you for writing a great paper.

To clarify the purpose of the study, it is recommended that you add a research background for PNI.

It seems to be very old data, not recent data. Is there a reason why you couldn't use the latest data? If necessary, describe the limitations in discussion sections and describe them as future studies.

Please describe IRB number in terms of research ethics.

I think it would be good to add T2N0 Oral Cancer Staining Figure in the material and method.

Please state the exact information about the SPSS.

The method of marking the P value requires unification, such as case and spacing.
p=0.25, p=0.0006 in tables and figures
In the body, p = 0.001

Please capitalize all the first letters of the word in the table.
e.g.) characteristics -> Characteristics

Please correct the spacing before the parentheses in the table.
e.g.) 28 (16.8) -> 28 (16.8)

Please capitalize the first letter of the word in the picture.
e.g.) positive -> Positive, months -> Months

Please check the reference number.

Please check the posting regulations for reference and correct the error.
e.g.) If the author's name is indicated, at least three people are et al. Journal name error, period writing error, etc

Author Response

Reviewer 1:

The main question was to review the data of patients of T2N0 OSCC to elucidate the risk factors of poor prognosis and to suggest possible management to improve the patient’s survival. The topic of originality is excellent and relevant in the field. The main question posed the evidence and arguments presented, the references are appropriate, and the tables and figures are also appropriate. 

However, there are some questions and comments, please check and modify if necessary.

Thank you for writing a great paper.

1.To clarify the purpose of the study, it is recommended that you add a research background for PNI.

Response: Chatzistefanou [9] reviewer previous articles and concluded that PNI is correlated with more aggressive tumor and poor outcomes. (page 2)

2.It seems to be very old data, not recent data. Is there a reason why you couldn't use the latest data? If necessary, describe the limitations in discussion sections and describe them as future studies.

Response: Different staging system may lead to the different results. In order to prevent this difference, patients who used the same staging system of the American Joint Committee on Cancer (AJCC) TNM classification system 7th edition (2012-2017) were included in this study.

3.Please describe IRB number in terms of research ethics.

Response: Institutional Review Board of Taichung Veterans General Hospital approved this retrospective study (protocol number CE19242A). (page 2)

4.I think it would be good to add T2N0 Oral Cancer Staining Figure in the material and method.

Response: Thank you for your valuable suggestion. It is difficult to get staining figure in this retrospective study.

5.Please state the exact information about the SPSS.

Response: SPSS software(version 12.0, IBM, NY). (page 3)

6.The method of marking the P value requires unification, such as case and spacing.
p=0.25, p=0.0006 in tables and figures
In the body, p = 0.001

Response: We corrected it.

7.Please capitalize all the first letters of the word in the table.
e.g.) characteristics -> Characteristics
Response: We corrected it. (Table 1)

8.Please correct the spacing before the parentheses in the table.
e.g.) 28 (16.8) -> 28 (16.8)

Response: We corrected it. (Table 1)

9.Please capitalize the first letter of the word in the picture.
e.g.) positive -> Positive, months -> Months
Response: We corrected it. (Figures)

10.Please check the reference number.

Response: We checked the reference number.

11.Please check the posting regulations for reference and correct the error.
e.g.) If the author's name is indicated, at least three people are et al. Journal name error, period writing error, etc

Response: We corrected it.

Reviewer 2 Report

comments have been added both here and in the attachment

The main objective of the study was to find the confounding factor which was most important in prognostication and treatment of T2NOMO OSCC. PNI has been clearly discussed in the paper. However, the author can add more literature pertaining to PNI and AVI. Also, the authors could have done a comparison among T3 and T4 with n0 m0 status to check for any differences in prognosis. However, this study could become a series of studies comparing different stages as well. English and grammatical corrections are required. The study can be accepted these minor corrections.

Author Response

Reviewer 2:

The main objective of the study was to find the confounding factor which was most important in prognostication and treatment of T2NOMO OSCC. PNI has been clearly discussed in the paper. However, the author can add more literature pertaining to PNI and AVI. Also, the authors could have done a comparison among T3 and T4 with n0 m0 status to check for any differences in prognosis. However, this study could become a series of studies comparing different stages as well. English and grammatical corrections are required. The study can be accepted these minor corrections.

Response: Thank you for your valuable suggestion. We corrected the manuscript as your suggestion. We will conduct the study focused on T3-4N0M0 oral cancer patients and to find out the importance of adverse features.
